# Lymphocystis Disease Virus Infection in Clownfish *Amphiprion ocellaris* and *Amphiprion clarkii* in Taiwan

**DOI:** 10.3390/ani13010153

**Published:** 2022-12-30

**Authors:** Ming-Chung Cheng, Ming She See, Pei-Chi Wang, Yu-Ting Kuo, Yuan-Shing Ho, Shih-Chu Chen, Ming-An Tsai

**Affiliations:** 1Eastern Marine Biology Center, Fisheries Research Institute, Taitung 961, Taiwan; 2Institute of Marine Biotechnology, Universiti Malaysia Terengganu, Kuala Nerus 21300, Terengganu, Malaysia; 3Department of Veterinary Medicine, College of Veterinary Medicine, National Pingtung University of Science and Technology, Pingtung 91201, Taiwan; 4International Program in Ornamental Fish Technology and Aquatic Animal Health, International College, National Pingtung University of Science and Technology, Pingtung 91201, Taiwan

**Keywords:** lymphocystis disease, virus, marine, clownfish, *Amphiprion ocellaris*, *Amphiprion clarkii*, PCR

## Abstract

**Simple Summary:**

The present research is about Lymphocystic Disease Virus (LCDV) infection in *Amphiprion ocellaris* and *Amphiprion clarkii*. Detection of LCDV depended on histopathological study, electron microscope observation of virus particles and gene sequence analysis from the MCP region. Symptoms of LCDV were sparse, multifocal, white, stiff, papilloma-like nodules on the body, skin, gills and fins. The viral particle present were virions 18–230 nm in diameter, hexagonal in shape with an inner dense nucleoid under transmission electron micrographs. The ML polygenetic tree constructed based on the MCP region of LCDV showed the LCDV genotype in the present study is closely related to LCDV from the paradise fish, *Macropodus opercularis* (KJ408271) (pairwise distance: 92.5%) from China. This is the first report of lymphocystis disease in *A. clarkii* in Taiwan.

**Abstract:**

Lymphocystic disease affects over 150 species of marine and freshwater fish worldwide. In this study, the lymphocystis pathogen was found in 2 (*Amphiprion ocellaris* and *Amphiprion clarkii*) of the 9 species of clownfish. Detection of lymphocystis disease virus (LCDV) was based on histopathological study, electron microscope observation of virus particles and gene sequence analysis from the MCP region. Infected *A. ocellaris* hosts showed sparse, multifocal, white, stiff, papilloma-like nodules on the body, skin, gills and fins; while, on *A. clarkia*, nodules were found on the operculum skin. Histopathologic study showed lymphocystic cells with an irregular nucleus, enlarged cytoplasm and intracytoplasmic inclusion bodies surrounded by the cell membrane. The viral particle presents virions 180–230 nm in diameter, hexagonal in shape with an inner dense nucleoid under transmission electron micrographs (TEM). From the ML polygenetic tree, the clownfish LCVD genotype was closely related to the LCDV strain from paradise fish, *Macropodus opercularis* (KJ408271) (pairwise distance: 92.5%) from China, then followed by the strain from Spain (GU320726 and GU320736) (pairwise distance: 90.8–90.5%), Korea (AB299163, AB212999, AB213004, and AB299164) (pairwise distance: 91.5–80.5%) and lastly Canada (GU939626) (pairwise distance: 83%). This is the first report of lymphocystis disease in *A. clarkii* in Taiwan.

## 1. Introduction

Lymphocystic disease has been discovered to affect over 150 species of marine and freshwater fish [1,2]. The appearance of this disease was initially described in the 19th century in which macroscopic nodules (0.3–2.0 mm) develop on the body surface and fins [1]. This is caused by the collection of hypertrophied dermal cells (lymphocystic cells) in the fish [3]. Lymphocystis disease virus (LCDV) belongs to the genus *Lymphocystivirus*, family *Iridoviridae*, one of the causative agents of lymphocystic disease. LCDV is a large (130–300 nm) double-stranded DNA virus, with a size range between 108 and 208 kilobase pairs (kbp) and, typically, with a low G+C content (27–33%) [4]. From a histopathological view, enlarged lymphocysts with a dermal fibroblast could reach sizes 100 times larger than the normal cells in fish hosts [5,6]. Previous studies also determined an icosahedral viral particle that measured 200 nm in diameter with a dense nucleocapsid and a well-defined electron-lucent envelope with surface-associated fibrils under transmission electron microscopy (TEM) [7,8]. Currently, 4 complete genome sequences have been isolated: LCDV-1 from flatfishes (*Platichtys flesus* L.) [9], LCDV-C from Japanese flounder (*Paralichthys olivaceus*) from China [10], LCDV-3 from gilthead sea bream (*Sparus aurata*) [6] and LCDV-WA from whitemouth croaker (*Micropogonias furnieri*) [11]. The major capsid protein (MCP) gene is commonly used for the phylogenetic studies of iridovirus because it is highly conserved within the family [12,13].

Unfavorable environmental conditions such as intensive fish farming, water contamination, nutritional deficiencies and oxygen depletion could increase the emergence of lymphocystic disease [14]. Exposure of skin lesions and damaged gill are targeted by the LCDV to access the host body [15]. In recent data, LCDV was found in high-value food fish: gilthead seabream, *Sparus aurata* [6]; senegalese sole, *Solea senegalensis* [12]; whitemouth croaker, *Micropogonias furnieri* [11]; as well as ornamental fish: Indian glassy fish, *Parambassis ranga* [16]; yellowbar angelfish, *Pomacanthus maculosus* [17]; and paradise fish, *Macropodus opercularis* [8]. Although lymphocystic disease is rarely fatal to fish, it may affect the fish host market value negatively through its unpleasant morphology, especially in the ornamental fish industry [7].

In the global ornamental fish market, clownfish are one of the most famous ornamental marine fish because of their unique color variety [18]. As the population of wild clownfish had dropped sharply due to man-made destruction and environmental pollution [19,20], artificial breeding and culture of clownfish are developed to fulfil the market demand. High intensity culture and the stressful environment of the farm lead to disease outbreaks. Infection of LCDV was observed in different species of clownfish: *Amphiprion ocellaris*, *A. pergola* [21] and *A. percula* in India [22] and *A. ocellaris* in Thailand [7].

In the present study, LCVD was found in a clownfish breeding farm in Taiwan. In addition to *A. ocellaris*, LCDV also caused infection in *A. clarkii*. Therefore, its characteristics were studied by histopathological study, electron microscope observation of virus particles and gene sequence analysis. The present research aims to study LCDV in two species of clownfish through histopathological changes, PCR and phylogenetic analysis. Our data were the first to provide new genotype molecular evidence of lymphocystis disease virus in clownfish. Further study on the virus morphogenesis and genome sequence annotation could provide new insights into understanding the pathogenetic mechanism of LCDV.

## 2. Materials and Methods

### 2.1. Collection of Fish Samples

The diseased clownfish, *A. ocellaris* (*n* = 7) and *A. clarkii* (*n* = 1), were collected from the Eastern Marine Biology Research Center, Fisheries Research Institute in Taitung, Taiwan. Broodfish were at least 2 years old when they were purchased, and they were kept for 2–3 years after purchase. There were 9 species of clownfish (*A. ocellaris*, *A. frenatus*, *A. latezonatus, Premnas biaculeatus*, *A. ocellaris var.*, *A. percula*, *A. clarkii*, *A. ephippium* and *Amphiprion* sp. cf. *clarkii*), but only *A. ocellaris* (14%, 7/50), and *A. clarkii* (10%, 1/10) had symptoms in the clownfish breeding room within 3 weeks, with no mortality. The diseased fishes were anesthetized in 0.008% MS 222 solution and then sacrificed, sampled and wet mounts of body surfaces, fins, and gills from the diseased fish were examined under a light microscope.

### 2.2. Pathology

Gross pathology, including infected location, distribution, size, shape, color, consistency and special features of typical external lesions, was done for all of the fish samples. Fish samples with lymphocystis nodules were selected and fixed with 10% buffered formalin solution at room temperature for 24 h and processed for paraffin sectioning. The sections were stained with hematoxylin and eosin (H&E).

### 2.3. Electron Microscopy

The irregular white nodules from the skin were fixed with 2.5% glutaraldehyde with 0.2 M sodium cacodylate and post-fixed in 1% osmium tetroxide in 0.2 M sodium cacodylate for transmission electron microscopy. Sections were then stained in uranyl acetate and lead citrate before examination.

### 2.4. PCR Detection of MCP Genes

Gill, skin, and spleen from diseased fish were collected and homogenized in phosphate-buffered saline (PBS, pH 7.2). The genomic DNA was prepared according to the phenol/chloroform extraction method of Cheng et al. [23].

MCP genes were amplified from diseased fish based on PCR with primers, MCPexp-F and MCPexp-R as described by Hossain et al. [24]. PCR was performed on a thermal cycler (Biorad, Hercules, CA, USA) with reaction mixtures containing 50 mM KCl, 10 mM Tris-HCl (pH 8.3), 1.5 mM MgCl2, 200 µM deoxynucleoside triphosphate, 10 µM primer, 4 µL (200 ng) of template DNA and 0.75 U of Blend Taq^®^ DNA polymerase (Toyobo, Osaka, Japan), to a final volume of 25 µL. Amplification conditions consisted of an initial denaturation step at 95 °C for 5 min, followed by 33 cycles of denaturation at 95 °C for 1 min, annealing at 54 °C for 1 min and extension at 72 °C for 1.5 min, except for the final cycle where the extension was for 10 min. Finally, the PCR amplified products (1390 bp) were resolved in 1% agarose gel by electrophoresis and purified with the Clean and Gel Extraction Kit (Biokit, Biotechnology, Inc., Catalonia, Spain).

### 2.5. Sequence Analysis

PCR amplicons from *A. ocellaris* and *A. clarkii* were sequenced by the Tri-I Biotech Company, New Tapei City, Taiwan. Initial comparisons with sequences in the GenBank Entrez databases were performed using BLAST (http://www.ncbi.nlm.nih.gov (accessed on 28 November 2022) to identify homologous sequences. The Bioedit Alignment Sequence Editor Ver. 7.0.5.3 was used to assemble and edit the forward and reverse sequences of MCP regions. The original chromatograms are compared manually if necessary. Obtained sequences were aligned with the previously characterized sequences of Iridovirus (Table 1) registered in GenBank by using Bioedit Alignment Sequence Editor Ver. 7.0.5.3 (Informer Technologies, Inc., Rheinland-Pfalz, Germany). The MEGA version 11 [25] was conducted for phylogenetic and molecular evolutionary analyses. Retrieved MCP region sequences are listed in Table 1. The maximum likelihood polygenetic tree and the pairwise distance were constructed by using MEGA 11 software by bootstrap analyses using 1000 resamplings.

## 3. Results

### 3.1. Macroscopic Observations

The fish samples showed sparse, multifocal, white, stiff, papilloma-like nodules on the body, skin, gills and fins in *A. ocellaris. A. clarkii* had only mild symptoms found on the operculum skin (Figure 1a,b).

### 3.2. Histopathology of Nodules from Diseased Clownfish

The gill and fin are wreathed by an unusually thick outer membrane, forming a hyaline capsule (Figure 2a,b). The hyaline capsule is shown with a thick and smooth layer covering the hypertrophic lymphocystis cells (Figure 3a,b). The hypertrophic cells are present in various sizes. In addition, the magnified hypertrophic lymphocystis cells showed an irregular nucleus, and intracytoplasmic inclusion bodies around the cell membrane were strongly stained by hematoxylin and eosin (H&E) (Figure 4a,b).

### 3.3. Electron Microscope

The existence of viral particles in the nodules of affected parts was observed by transmission electron micrographs (Figure 5a,b). Transmission electron microscopy (TEM) of the diseased fish shows large numbers of the virus particle (Figure 5a) in the cytoplasm of infected cells, with virions 180–230 nm in diameter, hexagonal in shape and with an inner dense nucleoid (Figure 5b).

### 3.4. Molecular Phylogeny

Five clownfish, *A. ocellaris* (*n* = 4) and *A. clarkii* (*n* = 1), were detected with LCDV infection through a genetic study. The gene sequencing from the MCP of five fish showed positive results in gel electrophoresis and had a 1390 bp length in their PCR product. Three of the selected organs of *A. ocellaris* showed positive signs of LCDV infection; while *A. clarkii* had mild symptoms with positive results in the gill and fin; LCDV was not detected in the spleen (Table 2).

After aligning all the sequences from *A. ocellaris* (*n* = 4) and *A. clarkii* (*n* = 1), the gene sequences were grouped into two different groups: AOD100112 (*A. clarkii*) and AOD100090-2A (*A. ocellaris*). A phylogenetic tree based on the MCP gene sequences of these two genes was constructed by the maximum-likelihood method using Molecular Evolutionary Genetics Analysis (MEGA 11). To further ascertain the phylogenetic relationship between the new genotype LCDV and other iridovirus isolates, another four iridovirus core genes, MCP, DNA polymerase, myristoylated membrane protein (MMP) and ribonucleotide reductase (RNR) were amplified. Phylogenetic trees were constructed using MEGA 11 based on the MCP gene sequences of these genes. As shown in Figure 6, both of the LCDV virus genes in this study showed 100% similarity and grouped into the same cluster. Similar results were also found from the LCDV found in Spain (accession numbers GU320726 and GU320736) that had 99.7% similarity. The phylogenetic analysis consistently revealed that the LCDV Taiwan cluster (AOD100112 and AOD100090-2A) shared the closest relationship to LCDV found in China (accession number KJ408271) (pairwise distance: 92.5%), followed by Spain (accession number GU320726 and GU320736) (pairwise distance: 90.8–90.5%), Korea (accession number AB212999, AB213004, AB299163 and AB299164) (pairwise distance: 91.5–80.5%), Canada (accession number GU939626) (pairwise distance: 83%) and other iridovirus isolates (pairwise distance: 57.3–55.7%) (Table 3). Taken together, we proposed that G-VII was a novel LCDV isolate in the family Iridoviridae.

## 4. Discussion

A unique LCDV strain was discovered in captive-bred clownfish in Taiwan, representing the first gene sequencing of LCDV in clownfish. Although LCDV was known to infect a wide range of fish hosts [1,2], cases in Taiwan were still undiscovered. In Asia, LCDV was commonly reported in high-value aquaculture fishes: flounder, *Paralichthys* spp. [26,34,35] and rockfish *Sebastes* spp. [28,36,37]. In addition, LCDV infection is frequently detected in farmed gilt-head bream, *Sparus aurata,* in Egypt [38], Spain [27] and Tunisia [39]. As LCDV could infect a wide range of species of fish [1,2], clownfish may also become a victim. Before the study of Lam et al. [22], the detection of lymphocystic disease had only detected by histopathology and electron microscopy [7,21]. The reported LCDV infection in clownfish were found to be in *A. ocellaris* [7,21] and *A. percula* [21,22]. In histopathological studies, each white nodule consists of one or more lymphocystic cells with an irregular nucleus, enlarged cytoplasm and intracytoplasmic inclusion bodies surrounding the cell membrane, which corresponds to the previous research in clownfish [21,22]. Numerous viral particles and the distinct structure of the viral particle could be found under TEM. The LCDV viral particle from *S. aurata* [38], *Solea senegalensis* [40], *Sebastes schlegelii* [37] and *A. ocellaris* [7] have a similar ultrastructure morphology (hexagonal in shape with an inner dense nucleoid). In the present study, 9 species of clownfish (*A. ocellaris*, *A. frenatus*, *A. latezonatus*, *Premnas biaculeatus*, *A. ocellaris var.*, *A. percula*, *A. clarkii*, *A. ephippium* and *Amphiprion* sp. cf. *clarkii*) were cultured in the same aquaculture system. However, only 8 fish from 2 species of clownfish, *A. ocellaris* (*n* = 7) and *A. clarkii* (*n* = 1), visually (white nodules on the external body structure: skin, fin and tail) and genetically showed LCDV infection. This is the first report demonstrating the presence of LCDV in *A. clarkii*. The infection was also confirmed by the degree of lesions and PCR. In recent studies, LCDV not only appears in the fin and skin of the fish host, but also internal organs such as the brain, liver, kidney, spleen and intestine [16,17,41]. The gene expression profile showed that the target organs of the LCDV are found to be the fin/skin, kidney, gut and liver [42,43,44,45]. In early infection, the LCDV enters the target organ from the bloodstream. Then, LCDV will spread to other organs during the virus outbreak or host immune suppression [42,43]. The present study showed both species of clownfish showed LCDV in the target organ (fin). However, LCDV was only found in the spleen of *A. ocellaris*, which indicates susceptibility to this virus is highest in *A. ocellaris*, followed by *A. clarkii*; the other species of clownfish are less sensitive. Moreover, the previous study showed the viral load was reduced progressively over the experiment infection time. The detection of the viral MCP gene transcription was detected in the kidney, intestine, skin-fin, liver and brain (except spleen) at 5 to 7 dpi, and only found in skin–caudal fin pools at 15 dpi [42]. This could be the reason for the mild symptoms of LCDV infection (no detection in the spleen) in *A. clarkii* which was sampled after 3 weeks of the LCDV outbreak.

Each genotype of LCDV is limited to a few or one species of fish, and infection of the same genotype of LCDV in different species of fish hosts is rare [12,13,28]. There is only one research study in Spain that found that LCDV genotype VII was able to cause infection in *S. aurata* (GU320726) and *S. senegalensis* (GU320736). The first detection of this LCDV genotype was found in cultured *S. aurata* and continued with infection in *S. senegalensis* after 3 months in the same fish farm. The LCDVs in genotype VII were found to have 99.7% similarity [12]. In the present study, the outbreak of LCDV in *A. ocellaris* and *A. clarkii* occurred in the same period, and the isolated MCP sequences of the two clownfish are 100% similar, representing the same iridovirus spreading in the system. This result could be similar to the infection of LCDV genotype VII, but the present study occurred in the same family of fish hosts and had different phenotypes of infection, even though the genetic relation of LCDV is more dependent on the species of host rather than geographical factors [28]. However, there is no full sequence of the MCP region of LCDV from clownfish in other research, and the ML polygenetic study showed that the genetic relationship of LCDV in the present research was related to geographical factors. The LCDV strains found in this study were closely related to the LCDV strains from paradise fish, *Macropodus opercularis* (KJ408271) (pairwise distance: 92.5%) from China, then followed by the strain from Spain (GU320726 and GU320736) (pairwise distance: 90.8–90.5%), Korea (AB299163, AB212999, AB213004 and AB299164) (pairwise distance: 91.5–80.5%) and lastly Canada (GU939626) (pairwise distance: 83%). Compared with the other 6 fish iridovirus genes, there was 57.3–55.7% similarity toward the isolated MCP sequence fen LCDV in the present study.

## 5. Conclusions

This is the first study that has provided new genotype molecular evidence of lymphocystis disease virus in clownfish, which might contribute greatly to the diagnosis and control of clownfish viral diseases. Although the vulnerability of fish to stress and disease could be enhanced by intensive breeding conditions, healthy management with good husbandry, early detection of disease pathogens and improved nutrition could eliminate disease outbreaks.

## Figures and Tables

**Figure 1 animals-13-00153-f001:**
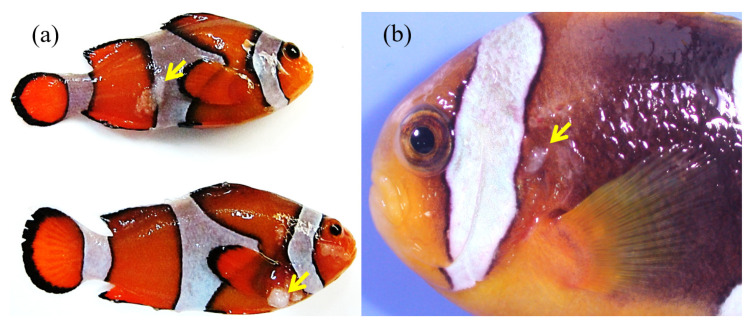
Macroscopic view of Lymphocystis disease virus (LCDV) infected *Amphiprion ocellaris* (**a**) and *A. clarkii* (**b**). Yellow arrows are white nodules.

**Figure 2 animals-13-00153-f002:**
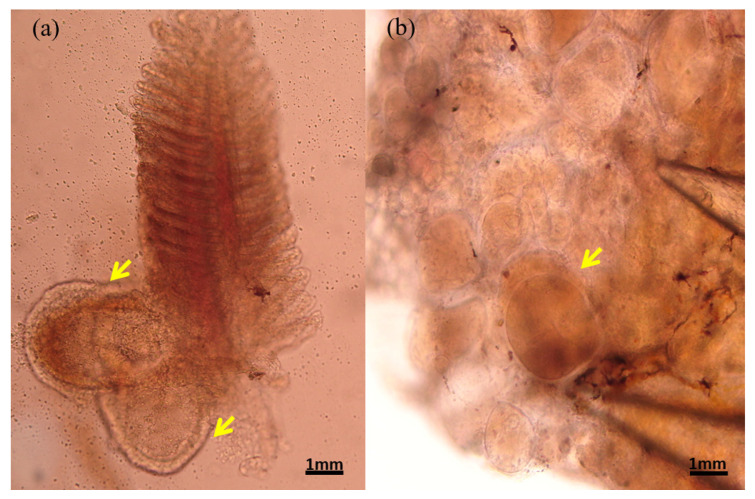
Macroscopic view of a 10× gill tissue slice (**a**) and a 10× fin tissue slice (**b**). Yellow arrow is a hyaline capsule.

**Figure 3 animals-13-00153-f003:**
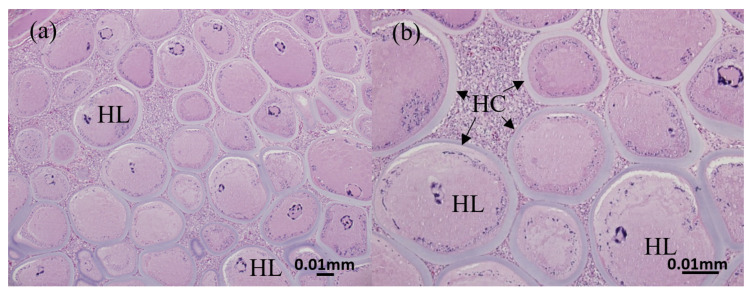
Histopathological findings of hypertrophic lymphocystis cells. (**a**) 40× view and (**b**) 40× view. HL: hypertrophic lymphocystis cell; HC with arrow: hyaline capsule.

**Figure 4 animals-13-00153-f004:**
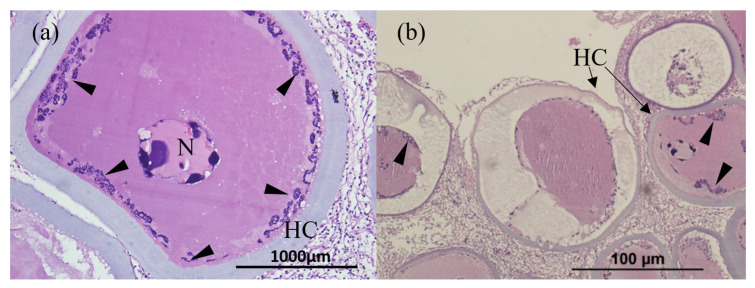
Histopathological findings of hypertrophic lymphocystis cells: (**a**) 100× view of single hypertrophic lymphocystic cell and (**b**) 100× view of lesion tissue on skin. HC: hyaline capsule; Arrowhead: intracytoplasmic inclusion bodies; N: nucleus.

**Figure 5 animals-13-00153-f005:**
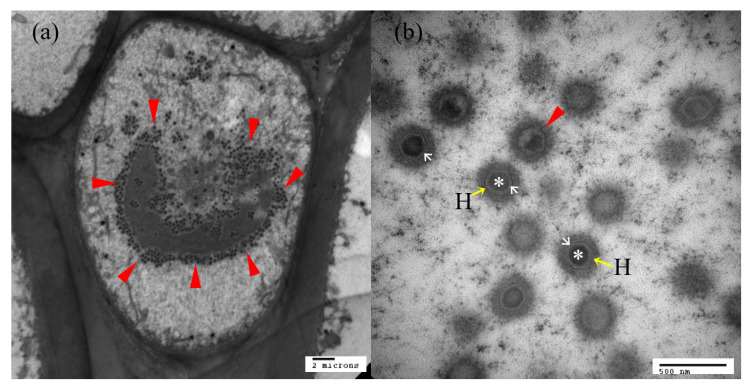
Transmission electron microscopy image of an ultra-thin section of a hypertrophic lymphocystis cell. (**a**) Ultrastructure of a hypertrophic lymphocystis cell with an LCDV virus particle, (**b**) LCDV virus. Red arrow: virus particle; H with yellow arrow: hexagonal profile protein shell; White arrow: inner membrane; *: central DNA core.

**Figure 6 animals-13-00153-f006:**
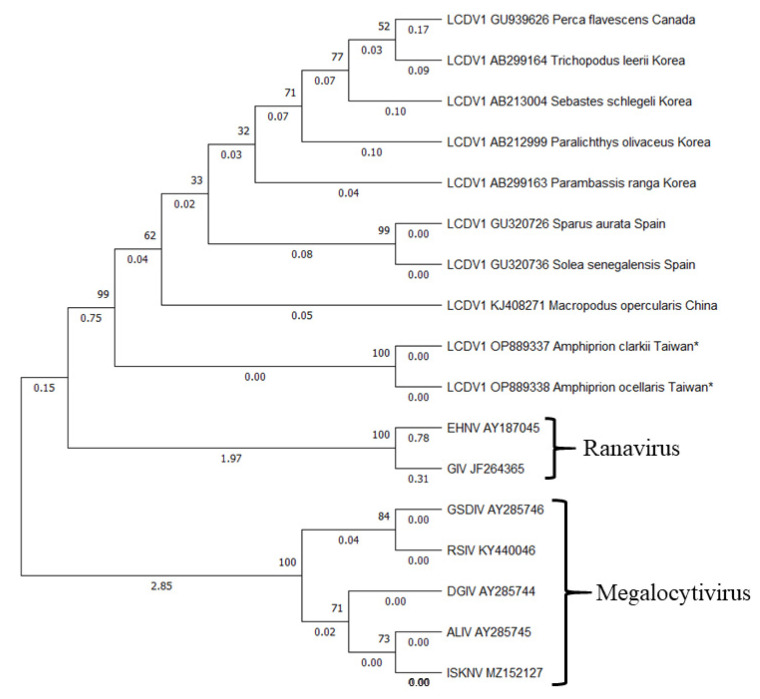
Maximum likelihood polygenetic trees from *Lymphocystivirus* spp. based on the major capsid protein (MCP) gene under the K2+G (Kimura 2 + Gamma Distributed) model. Accession numbers were reported in the text. Number on node is 1000 bootstrap replications. * represent the present study.

**Table 1 animals-13-00153-t001:** List of the MCP region sequences used in phylogenetic analysis.

Accession Code	Virus Species	Host	Region	Size	References
OP889337	LCDV 1	*Amphiprion clarkii*	Taiwan	1390	Present study
OP889338	LCDV 1	*Amphiprion ocellaris*	Taiwan	1390	Present study
AB299163	LCDV 1	*Parambassis ranga*	Korea	1337	[26]
KJ408271	LCDV 1	*Macropodus opercularis*	China	1208	[8]
GU320726	LCDV 1	*Sparus aurata*	Spain	1317	[12]
GU320736	LCDV 1	*Solea senegalensis*	Spain	1317	[27]
AB212999	LCDV 1	*Paralichthys olivaceus*	Korea	1347	[28]
GU939626	LCDV 1	*Perca flavescens*	Canada	1357	[13]
AB299164	LCDV 1	*Trichopodus leerii*	Korea	1337	[24]
AB213004	LCDV 1	*Sebastes schlegeli*	Korea	1356	[28]
JF264365	GIV	*Epinephelus coioides*	Taiwan	1392	[29]
AY187045	EHNV	*Perca fluviatilis*	Australia	1472	[30]
AY285746	GSDIV	*Epinephelus chlorostigma*	Thailand	1362	[31]
KY440046	RSIV	*Lates calcarifer*	Vietnam	1362	[32]
AY285745	ALIV	*Aplocheilichthys centralis*	Indonesia	1362	[31]
MZ152127	ISKNV	*Osphronemus goramy*	India	1362	[33]
AY285744	DGIV	*Colisa lalia*	Malaysia	1362	[31]

LCDV 1, Lymphocystis disease virus 1; GIV, Grouper iridovirus; EHNV, hematopoietic necrosis virus; GSDIV, Grouper sleepy disease iridovirus; RSIV, Red seabream iridovirus; ALIV, African lampeye iridovirus; ISKNV, Infectious spleen and kidney necrosis virus; DGIV, Dwarf gourami iridovirus.

**Table 2 animals-13-00153-t002:** Detection of the Lymphocystis disease virus (LCDV) major capsid protein (MCP) gene from different organs of clownfish, *Amphiprion* spp.

Clownfish Species	Organ		
	Gill	Fin	Spleen
*Amphiprion clarkii* (*n* = 1)	+	+	-
*Amphiprion ocellaris* (*n* = 4)	+	+	+

**Table 3 animals-13-00153-t003:** Maximum likelihood polygenetic trees from *Lymphocystivirus* spp. based on the major capsid protein (MCP) gene under the K2+G (Kimura 2 + Gamma Distributed) model. Accession numbers were reported in the text. Number on node is 1000 bootstrap replications.

No.	Accession Number	1	2	3	4	5	6	7	8	9	10	11	12	13	14	15	16
1	OP889337/ OP889338	100.0	92.5	91.5	90.8	90.5	83.4	83.0	81.3	80.5	57.3	55.9	55.9	55.9	55.9	55.7	55.7
2	KJ408271		100.0	89.9	89.6	89.2	86.0	83.8	83.9	83.3	56.6	52.8	54.4	54.2	54.2	54.1	54.2
3	AB299163			100.0	89.1	88.8	86.1	82.9	84.3	85.2	55.7	52.2	54.2	53.7	53.5	53.5	54.1
4	GU320726				100.0	99.7	84.8	83.4	83.2	84.0	56.1	53.6	54.8	54.1	54.1	54.1	54.7
5	GU320736					100.0	84.5	83.1	82.9	83.7	55.9	53.6	54.7	54.0	54.0	54.0	54.5
6	AB212999						100.0	82.3	84.8	85.6	56.3	50.8	52.2	51.9	51.7	51.6	52.2
7	GU939626							100	84.6	83.5	55.8	51.0	53.8	53.3	53.3	53.1	53.7
8	AB299164								100.0	87.4	54.9	49.2	52.7	52.3	52.2	52.1	52.7
9	AB213004									100.0	54.3	49.9	52.4	51.1	50.9	50.9	52.2
10	GIV										100.0	69.9	50.7	50.7	50.9	50.5	50.5
11	EHNV											100.0	54.6	54.5	54.5	54.3	54.7
12	RSIV												100.0	95.0	94.6	94.9	99.4
13	ALIV													100.0	99.7	99.7	94.7
14	ISKNV														100.0	99.3	94.4
15	DGIV															100.0	94.6
16	GSDIV																100.0

## Data Availability

Not applicable.

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
