# Peer review of "Lymphocystis Disease Virus Infection in Clownfish Amphiprion ocellaris and Amphiprion clarkii in Taiwan"

_animals, 2022, doi:10.3390/ani13010153_

Round 1
Reviewer 1 Report
In this study, infection of Lymphocystic Disease Virus (LCDV) in Amphiprion Ocellaris and Amphiprion Clarkii were reported. Detecion of LCDV was depended on the histopathological study, electron microscope observation of virus particles and gene sequence analysis from MCP region. The results extended the data about the host range of LCDV and provide better understanding of Lymphocystic Disease. It is can be published in Animals after some issues are resolved.
---Line 237: “were culture un a same aquaculture system” should be “in a same aquaculture system””
---Line 242-243: “LCDV not only appear in the fin and skin of the fish host, internal organ such as brain, liver, kidney, spleen and intestine”. “but” should be added before “internal organ”
---The description “Gene expression profile showed the target organ of the LCDV are found to be the fin and kidney [36-38]” was inaccurate, because the target organ of the LCDV are not only the fin and kidney.
---Line 248: “the LCDV only found int the spleen”, should be “in the spleen”
---In Abstract and Discussion, the description “the clownfish LCVD genotype was close related to the LCDV strain from paradise fish, Macropodus opercularis (KJ408271) (Pairwise distance: 92.5%) from China, then follow by the strain from Spain (GU320726 and GU320736) (Pairwise distance: 90.8-90.5%), Korea (AB299163, AB212999, AB213004 and AB299164) (Pairwise distance: 91.5-80.5%) and lastly Canada (GU939626) (Pairwise distance: 83%).” The virus strain in different country is from different fish species, just mentioning the country is not suitable, it is recommended to describe the fish species but not country. Actually, it is possible that several LCDV strains may exist in a country.
---line 241, 250 “A. clarkia” should be in italics.
--- line 245: “the LCDV enter the target organ from the blood stream”, Reference might be provided. A recent study may provide evidence “Sheng, X.Z.; Zeng, J.; Zhong, Y.; Tang, X.Q.; Xing, J.; Chi, H.; Zhan, W.B. Peripheral Blood B-Lymphocytes Are Involved in Lymphocystis Disease Virus Infection in Flounder (Paralichthys olivaceus) via Cellular Receptor-Mediated Mechanism. Int. J. Mol. Sci. 2022, 23, 9225.”
Author Response
Thanks for the nice comments from the reviewer. We replied your comments one by one as following.
- Line 237: “were culture un a same aquaculture system” should be “in a same aquaculture system””.
Has been changed the word “in” at the sentence (Line 237).
- Line 242-243: “LCDV not only appear in the fin and skin of the fish host, internal organ such as brain, liver, kidney, spleen and intestine”. “but” should be added before “internal organ”.
According to the comment, we added the word “but” before “internal organ” (Line 242).
- The description “Gene expression profile showed the target organ of the LCDV are found to be the fin and kidney [36-38]” was inaccurate, because the target organ of the LCDV are not only the fin and kidney.
According to the comment, we revised Gene expression profile showed the target organ of the LCDV are found to be the “fin/skin and internal organs such as kidney, gut and liver “[36-39] at the sentence (Lines 243-244).
- Line 248: “the LCDV only found int the spleen”, should be “in the spleen”
Has been revised and please refer to Line 248
- In Abstract and Discussion, the description “the clownfish LCVD genotype was close related to the LCDV strain from paradise fish, Macropodus opercularis (KJ408271) (Pairwise distance: 92.5%) from China, then follow by the strain from Spain (GU320726 and GU320736) (Pairwise distance: 90.8-90.5%), Korea (AB299163, AB212999, AB213004 and AB299164) (Pairwise distance: 91.5-80.5%) and lastly Canada (GU939626) (Pairwise distance: 83%).” The virus strain in different country is from different fish species, just mentioning the country is not suitable, it is recommended to describe the fish species but not country. Actually, it is possible that several LCDV strains may exist in a country.
Limited by should be a total of about 200 words maximum in abstract. The information of fish species and country in LCVD infection has described in Table 1 and figure 6.
- line 241, 250 “A. clarkia” should be in italics
Has been revised and please refer to lines 240, 250
- line 245: “the LCDV enter the target organ from the blood stream”, Reference might be provided. A recent study may provide evidence “Sheng, X.Z.; Zeng, J.; Zhong, Y.; Tang, X.Q.; Xing, J.; Chi, H.; Zhan, W.B. Peripheral Blood B-Lymphocytes Are Involved in Lymphocystis Disease Virus Infection in Flounder (Paralichthys olivaceus) via Cellular Receptor-Mediated Mechanism. Int. J. Mol. Sci. 2022, 23, 9225.”
Has been cited and please refer to lines 245, 376-378
Reviewer 2 Report
In the article "Lymphocystis Disease Virus Infection in Clownfish Amphiprion ocellaris and Amphiprion clarkii in Taiwan" the authors describe an experiment on clownfish and characterize the virus, which causes nodules on gills, fin and internal organs.
The article is written in an understandable way. The introduction provides necessary information about LCD virus and informs what will be presented in this article.
The material and methods part describes what was done during the experiment, like: electron microscopy, molecular biology.
All tables and figures are readable and put in the right place in the text. Results are presented very clearly and understandably.
The discussion refers to other references and refers to obtained results.
I think this is a valuable article which should be published, but I recommend several minor corrections:
line 105 - shoudn't be Electron Microscopy?
line 221 - Asia, not Aisa
line 224, 225 - Egypt or Tunisia are not European countries
line 237 - in, not un
line 238, 239 - A. ocellaris and A. clarkia - write in italic
line 247 - in brackets should be only numbers from references
Author Response
Thanks for the nice comments from the reviewer. We replied your comments one by one as following.
- line 105 - shoudn't be Electron Microscopy?
Has been changed the word “Electron Microscopy” at the sentence (line 105)
- line 221 - Asia, not Aisa.
Has been revised and please refer to line 221
- line 224, 225 - Egypt or Tunisia are not European countries.
Has been removed “European countries” and please refer to line 224
- line 237 - in, not un
Has been changed the word “in” at the sentence (Line 237).
- line 238, 239 - A. ocellaris and A. clarkia - write in italic
Has been revised and please refer to line 238
- line 247 - in brackets should be only numbers from references
Has been revised and please refer to line 247
Reviewer 3 Report
In this study, lymphocystis pathogen was found in 2 of the 9 species of clownfish (Amphiprion 25
ocellaris and Amphiprion clarkii). This is the first report of lymphocystis disease in A. clarkia in Taiwan.
1. There are many errors in grammar and syntax throughout the text of the manuscript, which are required for correction, such as 64-65.
2. Line 183-185, Three of the selected organ of A. ocellaris were showing positive signal of LCDV infection; while A. clarkii with mild symptom was showing positive result in gill and fin, LCDV was no detected in the spleen (Table 2). The spleen is an important immune organ of fish.Why was LCDV not detected in the spleen? What are the target organs of LCDV infection?
Author Response
Thanks for the nice comments from the reviewer. We replied your comments one by one as following.
1. There are many errors in grammar and syntax throughout the text of the manuscript, which are required for correction, such as 64-65.
Has been revised and please refer to lines 63-64 and others (see Track Changes in text)
2. Line 183-185, Three of the selected organ of A. ocellaris were showing positive signal of LCDV infection; while A. clarkii with mild symptom was showing positive result in gill and fin, LCDV was no detected in the spleen (Table 2). The spleen is an important immune organ of fish.Why was LCDV not detected in the spleen? What are the target organs of LCDV infection?
The previous study showed the viral load was reduced progressively through the experiment infection time. The detection of viral MCP gene transcription was detected in the kidney, intestine, skin-fin, liver and brain (except spleen) at 5 to 7 dpi, and only found in skin-caudal fin pools at 15 dpi [36]. This could be the reason for mild symptom of LCDV infection (no detection in the spleen) in A. clarkia which was sampled after 3 weeks of the LCDV outbreak. (lines 249-255)